# On the Roles of the Nuclear Non-Coding RNA-Dependent Membrane-Less Organelles in the Cellular Stress Response

**DOI:** 10.3390/ijms24098108

**Published:** 2023-04-30

**Authors:** Anastasia A. Gavrilova, Anna S. Fefilova, Innokentii E. Vishnyakov, Irina M. Kuznetsova, Konstantin K. Turoverov, Vladimir N. Uversky, Alexander V. Fonin

**Affiliations:** 1Laboratory of Structural Dynamics, Stability and Folding of Proteins, Institute of Cytology, Russian Academy of Sciences, 194064 St. Petersburg, Russia; asultanbekova@incras.ru (A.A.G.); a.fefilova@incras.ru (A.S.F.); imk@incras.ru (I.M.K.); kkt@incras.ru (K.K.T.); 2Group of Molecular Cytology of Prokaryotes and Bacterial Invasion, Institute of Cytology, Russian Academy of Sciences, 194064 St. Petersburg, Russia; innvish@incras.ru; 3Department of Molecular Medicine and USF Health Byrd Alzheimer’s Research Institute, Morsani College of Medicine, University of South Florida, Tampa, FL 33612, USA

**Keywords:** LLPS, non-coding RNA, intrinsically disordered proteins, membrane-less organelles, stress response

## Abstract

At the beginning of the 21st century, it became obvious that radical changes had taken place in the concept of living matter and, in particular, in the concept of the organization of intracellular space. The accumulated data testify to the essential importance of phase transitions of biopolymers (first of all, intrinsically disordered proteins and RNA) in the spatiotemporal organization of the intracellular space. Of particular interest is the stress-induced reorganization of the intracellular space. Examples of organelles formed in response to stress are nuclear A-bodies and nuclear stress bodies. The formation of these organelles is based on liquid–liquid phase separation (LLPS) of intrinsically disordered proteins (IDPs) and non-coding RNA. Despite their overlapping composition and similar mechanism of formation, these organelles have different functional activities and physical properties. In this review, we will focus our attention on these membrane-less organelles (MLOs) and describe their functions, structure, and mechanism of formation.

## 1. Introduction

In 1924, Alexander I. Oparin (1894–1980) proposed a model suggesting that the formation of liquid coacervates due to liquid–liquid phase separations represented a crucial step in the origin of primitive life [1]. The theory was based on the observation that droplets of organic molecules coalesce spontaneously in solution. Unfortunately, this model was ultimately not supported by the scientific community because it could not explain the existence of membrane-bound organelles and their contribution to cellular compartmentalization.

However, the so-called “LLPS-revolution” that took place in the first decade of the 21st century radically changed the idea of the spatial-temporal organization of the intracellular space in line with Oparin’s view. It became obvious that the LLPS (liquid-liquid phase separation) of biopolymers plays an important role in the formation, regulation, and functional activity of cellular compartments. An entire class of organelles being formed as a result of a tightly controlled LLPS and not enclosed by the membranes (membrane-less organelles, MLOs) has been separated.

Membrane-less organelles are present in eukaryotes, Archaea, and bacteria (and, likely in viruses as well). They can be formed in the nucleus, in the cytoplasm, and in the mitochondrial matrix and chloroplast stroma. To date, about a hundred different MLOs are known [2]. Major nuclear MLOs are nucleoli, Cajal bodies, nuclear speckles, paraspeckles, nuclear stress bodies, amyloid bodies (A-bodies), and others. The cytoplasm also contains a number of MLOs, including processing bodies (P-bodies), stress granules, and germ granules [3,4]. While some MLOs are found in the cell under normal conditions (e.g., nucleoli, Cajal bodies, nuclear speckles, paraspeckles, histone loci bodies, P-bodies, etc.), others are formed under stress conditions.

One of the vital foundations for the existence and development of cells is the ability of living organisms to adapt to changes in the environment. To survive, cells must constantly regulate the multitude of cascades of biochemical reactions in response to intra- and extracellular signals. The ability of cells to form assemblages of proteins and nucleic acids in response to stress seems to be a conserved mechanism that allows them to withstand adverse conditions. Many cellular membrane-less organelles are condensates of proteins and nucleic acids. Due to the reversibility of biopolymer liquid-liquid phase transitions, the formation and disassembly of such structures can be extremely fast [5,6], which ensures rapid metabolic and functional restructuring and response to stress [7,8]. Examples of such MLOs are nuclear stress bodies and A-bodies.

Nuclear stress bodies (nSBs) and amyloid bodies (A-bodies or ABs) are two types of MLOs that are temporarily formed in the nucleus in response to stress and disintegrate during the period of post-stress recovery. Both of these stress-induced MLOs have a similar mechanism of assembly and play a number of significant roles in the activation of survival programs and the normalization of cellular homeostasis after the cell exits unfavorable conditions. In this review, we will focus our attention on these MLOs and describe their functions, structure, and mechanism of formation.

## 2. Nucleolar Stress Bodies

Nuclear stress bodies (nSBs) were first discovered in the late 1980s when it became clear that these structures are associated with the cellular response to stress [9]. nSBs are primate-specific MLOs that arise in response to various types of stress. The size and number of nSBs depend on the type and duration of stress; however, on average, one cell might contain several nSBs of 1–2 μm in size [6]. These organelles play an important role in the regulation of gene expression and the recovery of metabolic processes after stress exposure. However, specific functions, as well as the molecular mechanisms of the formation and decomposition of these compartments, remain poorly understood to date.

### 2.1. Structure and Formation Mechanism of nSBs

The assembly of nSBs is initiated by the transcription of highly repetitive satellite III architectural noncoding RNAs, HSATIII arcRNAs, or simply HSATIII, from the pericentromeric HSATIII repeat arrays of chromosomes 9, 12, and 15. However, nSBs are mainly associated with chromosome 9 (locus 9q12). HSATIII RNA is specific for primates; no formation of nuclear stress bodies or structures similar to them is observed in rodent cells in response to stress [9]. However, it is known that in Drosophila melanogaster, so-called omega-speckles are formed in response to temperature stress. They are nuclear structures and contain a long non-coding RNA (lncRNA), Hsrω, and many RNA-binding proteins, including various heterogeneous nuclear ribonucleoproteins (hnRNPs). Presumably, omega speckles are functional analogs of nSBs that arose independently in the course of evolution, which may emphasize the significance of such structures [10].

The heat shock response (HSR) activates not only the transcription of genes encoding heat shock proteins (HSP) [10] but also the transcription of pericentric heterochromatin. The pericentric heterochromatin is formed by tandem repeats of satellite III elements, which have a modular structure consisting of the GGAAT pentamer sequences separated by the “terminator” sequence CAAC(C/A)CGAGT [11]. Transcription activation is under the control of heat shock factor 1 (HSF1), a transcription factor present in the nucleus and cytoplasm of cells under normal conditions [12,13]. HSF2 is also involved in transcription activation [14]. Although vertebrate cells express five additional members of the HSF family (HSF3, HSF4, HSF5, HSFX, and HSFY), the actual roles of these proteins are mostly unknown [14].

Activation of pericentric heterochromatin is mainly observed at the 9q12 locus and leads to the accumulation of long non-coding (lnc) satellite III (sat III) RNA, which serves as a focus for the formation of nSBs [15]. As a result of thermal exposure of the cell, HSF1 recruits HAT histone acetyltransferases, such as general control of amino acid synthesis protein 5 (GCN5, also known as lysine acetyltransferase 2 (KAT2)), 60 kDa Tat-interactive protein (TIP60, also known as KAT5), and p300 to the region of pericentric heterochromatin, which leads to a redistribution of histone acetylation. The acetylated lysine residues of H3 and H4 histones then serve as docking sites for the bromodomain proteins of the BET family, triggering chromatin remodeling, RNAP II recruitment, and activation of HSATIII transcription [16]. Subsequently, the transcribed lncRNA HSATIII remains associated with its transcription sites (Figure 1).

The peak of SatIII RNA expression is observed under cell stress (heat shock) at 42–43 °C. While lower temperatures (39–40 °C) or high stress conditions (45 °C) have almost no effect on SatIII DNA expression, heat shock at 41 °C or 44 °C leads to ∼10% of maximal transcription [17]. Under hyperosmotic stress, the transcriptional activation of SatIII DNA involves the transcription factor tonicity enhancer binding protein (TonEBP, also known as nuclear factor of activated T-cells 5, NFAT5) [17]. TonEBP/NFAT5 is known to regulate gene expression in response to osmotic stress and is vital for kidney function and protection against elevated salt and urea levels in the renal medulla. Knockdown of TonEBP/NFAT5 with siRNA prevents nSB formation in response to stress, and the HSATIII sequence contains a putative TonEBP/NFAT5 binding site. Therefore, HSATIII RNA is a noncoding RNA whose transcription is controlled by different transcription factors under various stresses, and nSBs are an important part of the overall cellular response to stress, responding to various stress-sensitive pathways depending on the type of exposure [17].

Cadmium sulfate and UV are also able to induce transcription of HSATIII RNA. Long-term exposure to cadmium sulfate is comparable to heat shock both in terms of the proportion of cells with nSBs (>90%), the number of nSBs per cell, and their sizes [17]. UV-irradiation of cells induces the formation of a smaller proportion of cells with nSBs (30–40%), and the sizes of these structures are usually smaller than the heat shock effect. Interestingly, in all the cases described above, transcription mainly proceeds from the G-rich chain. C-rich transcripts are practically not found in cells not subjected to stress, and slightly increase after stress. Thus, the transcription of two DNA strands of HSATIII is asymmetric [17].

In a study [18], the authors induced knockdown of normal HSATIII expression using transient transfection with antisense oligonucleotides and found that knockdown significantly reduced (~70%) the survival of HeLa cells after thermal exposure compared to the cells that were transiently transfected with control oligonucleotides and subjected to heat stress. They then transiently transfected the mammalian expression construct (pcDNA-Sat3) to allow constitutive transcription of HSATIII under normal (stress-free) conditions. Cells that constitutively expressed the cloned HSATIII repeats exhibited significantly lower survival after heat shock exposure compared to those transfected with a blank control plasmid. These data indicate not only the importance of HSATIII expression during thermal stress but also the toxic effect of HSATIII transcripts during ectopic expression [18].

It is known that many proteins are involved in the biogenesis of nSBs. A recent proteomic analysis of isolated nSBs complexes has identified more than 140 proteins, mostly RNA-binding proteins (RBPs) that are involved in mRNA splicing, processing, and export [19]. SATIII transcripts are known to colocalize with HSF1, HSF2, scaffold attachment factor B (SAFB), splicing factor CREBBP (cyclic adenosine monophosphate response element (CREB)-binding protein) [18], serine/arginine-rich splicing factors 1 and 9 (SRSF1 and SRSF9) [19,20,21], and many other proteins. 

#### 2.1.1. HSF1

HSF1 is the most studied heat shock transcription factor and is the classic HSF that responds to elevated temperatures and other forms of stress. Under normal physiological conditions, HSF1 exists in an inactive state associated with molecular chaperones. Under stress, it is rapidly released from the complex, trimerized, hyperphosphorylated, and acquires DNA-binding and transcriptional activity [9,14,22]. Under constant moderate heat stress, the equilibrium shifts back to a monomeric, dephosphorylated, and Hsp-bound state. Interestingly, long-term stress leads to a decrease in HSF1 activity, which leads to a suppression of the ability to reactivate HSF1 during subsequent stress. A similar repression of HSF1 activity is observed during cell recovery after stress conditions. This acquired adaptation of cells to stressful influences is called stress thermotolerance [22].

#### 2.1.2. HSF2

Unlike HSF1, HSF2 undergoes a dimer-to-trimer transition upon activation and is not phosphorylated [13]. The regulation of HSF1 and HSF2 expression differs sharply. Unlike stably and constitutively expressed HSF1, HSF2 is inducible. The study [20] showed that HSF1 and HSF2 are functionally related. Heat shock produces HSF1 homotrimers and HSF1-HSF2-HSF1 heterotrimers, which activate and regulate HSATIII transcription in response to stress. The heterotrimerization mechanism of HSF1 and HSF2 likely provides transient regulation as heat shock lowers HSF2 levels, thereby limiting heterotrimerization due to limited HSF2 availability. Therefore, HSF1-HSF2 heterotrimerization regulates stress-induced transcription [20].

#### 2.1.3. SAFB

SAFB is an hnRNP protein that binds to already-formed stress granules via an RNA-binding domain [21]. It is a protein of the nuclear matrix and is involved in such important cellular processes as Xist-regulated inactivation of the X chromosome [21] and cell apoptosis [23]. SAFB supports the organization of pericentromeric heterochromatin, interacts with repeat element RNAs, and drives phase separation. It is known that the depletion of SAFB leads to structural changes in the organization of the genome, affecting the expression of various genes [24]. Therefore, the recruitment of SAFB into nSBs leads to the remodeling of genome compartmentalization, which probably helps the cell survive under stress [25].

At the very beginning of the stress response, the main protein in nSBs is HSF1, but its amount rapidly decreases during the recovery period after the cessation of the stimulus. The binding of SAFB to nSBs begins an hour after exposure to mild thermal stress, reaching a maximum concentration 3 h after the end of the stress. In their 2019 article, T. Hirose et al. suggested that in addition to the canonical HSF1/SAFB nSBs, additional stress nuclear bodies are also formed in the cell as a result of the interaction of the hnRNPM protein and HSATIII [26]. Whether hnRNPM forms a specific, distinct type of nSBs, or whether these formations represent one of the stages in the formation of nuclear stress bodies, remains to be seen.

#### 2.1.4. SRSF1 and SRSF9

SRSF1 and SRSF9 are pre-mRNA splicing factors capable of interacting simultaneously with RNA and other protein components through the RNA recognition motif (RRM) and through the RS domain rich in arginine and serine residues [19]. SRSF functions in pre-mRNA splicing are regulated by phosphorylation of the RS domain. Splicing factors (SRSF) are known to be rapidly dephosphorylated when exposed to thermal stress. During stress recovery, CDC-like kinase 1 (CLK1) is recruited to nSBs and re-phosphorylates SRSF9, which, by promoting intron retention, is involved in splicing inhibition [27]. In other words, nSBs serve as a platform for the rapid rephosphorylation of certain SRSFs and promote intron retention in hundreds of pre-mRNAs. There is another type of splicing regulation by nSBs that has been described. In a study [28], it has been shown that lncRNA HSATIII m^6^ A-methylated in the GGAAU repeat, which mediates YTHDC1 (YTH domain-containing protein 1) sequestration to nSBs during recovery from heat stress. This leads to the repression of m^6^ A-dependent splicing of specific pre-mRNAs.

The two mechanisms described above work in parallel to regulate splicing of target introns, with different features being recognized: a lower GC content and longer exon upstream for the YTHDC1 pathway and a relatively longer exon downstream for the SRSF9 pathway. However, at least some HSATIII target introns, such as RBM48 intron 3, are co-regulated by both mechanisms [19,28].

#### 2.1.5. CREBBP

CREBBP is another important protein that is part of nSBs. It is a transcriptional coactivator that forms a complex with various transcription factors and promotes the location of these complexes near the promoters of target genes. Recruitment of CREBBP to nSBs during heat shock occurs in an SRSF-1-dependent manner and contributes to the regulation of transcription under stress [18].

In 2020, a study was published proving that nSBs are dynamic molecular condensates formed as a result of liquid–liquid phase separation [29]. Furthermore, it has been shown that with an increase in the duration of stress exposure, the dynamics of HSF1 exchange with the nucleoplasm decreases sharply, which indicates a gradual hardening of HSF1-nSBs. Furthermore, after 16 h of incubation with the proteasome inhibitor MG132, HSF1 mobility decreased by 40%, which was also confirmed by the addition of the liquid droplet condensate inhibitor 1,6-hexanediol, showing a decrease in the mobile fraction by 54% after 8 h of proteotoxic exposure [29]. The dynamism and reversibility of the nSB formation promote cell survival. The appearance of insoluble gel-like HSF1-nSBs leads to an increase in cell sensitivity to apoptosis [29].

### 2.2. Biological Functions of nSBs

At present, the biological function of nSBs is not fully understood. Initially, it was assumed that nSBs are the transcription centers of HSF1 target genes, but this assumption was refuted when it was unambiguously shown that the localization of nSBs does not coincide with the centers of expression of heat shock proteins [28]. In addition, nSBs do not contain poly(A)-containing RNAs [18]. However, it is clear that activation of HSATIII transcription in cells exposed to heat shock remodels gene expression on a genome-wide scale due to massive recruitment of HAT and transcription factors to nSBs [16,19,20,28].

In splicing control, nSBs serve a dual function: as “molecular sponges” for YTHDC1 sequestration, and as “reaction crucibles” for SRSF phosphorylation. Both mechanisms contribute to the repression of pre-mRNA splicing during recovery from heat stress [20,28].

Therefore, the formation of nSBs in response to stress ensures the occurrence of cytoprotective reactions and, probably, is of great importance in the cell life cycle.

## 3. A-Bodies

It is known that both the synthesis and biogenesis of ribosomes and the processing of ribosomal RNAs take place in the nucleolus [30]. The nucleolus is a phase-separated biomolecular condensate in which a dynamic exchange of proteins with the surrounding nucleoplasm takes place [31]. Each nucleolus is organized in layers that are clearly visible under electron microscopy (EM): fibrillar center (FC); dense fibrillar component (DFC) enriched in fibrillarin (FIB1); and a granular component (GC) containing nucleophosmin (Figure 2) [31]. 

Heat-shocked MCF7 cells were fixed with 50% glutaraldehyde (Ted Pella, Redding, CA, USA) added to the cell suspension in the growth medium to a final concentration of 2.5% for 30 min. The cells were collected by centrifugation for 5 min at 1000 rpm. The supernatant was discarded. The pellet was treated with 1% OsO_4_ (30 min), dehydrated with increasing ethanol concentrations (70, 96%) and 100% acetone, impregnated, embedded into the Araldite M mixture (EMbed812, EMS, USA), and polymerized for 2 days at 60 °C. Ultrathin sections were prepared with a Ultratome III 8800 microtome (LKB, Stockholm, Sweden) and contrasted with Uranyl Acetate Alternative (Ted Pella, Redding, CA, USA) for 15 min and visualized under a Libra 120 electron microscope (Carl Zeiss, Jena, Germany) at magnification 8000×.

It should be noted that such a multilayer structure is not unique to the nucleolus. For example, stress granules and other RNP fluid-like bodies exhibit a similar core–shell structure [32,33]. The transcription process occurs at the FC/DFC interface, the early stages of co-transcriptional processing and modification of pre-rRNA occur in the DFC, and the late stages of pre-rRNA processing and assembly with ribosomal proteins occur in peripheral GCs [34,35].

The nucleolus is the largest and most easily detectable MLO. They are assembled by the phase separation of their molecular components. A study [36] showed that subcompartments within the nucleolus are separate, coexisting liquid phases. The layered organization of the droplets is caused by differences in the biophysical properties of the phases, especially in the surface tension of the droplets, which arise due to the peculiarities of their macromolecular components. Therefore, there is a molecular mechanism by which intrinsically disordered regions (IDRs) facilitate the condensation of the protein into droplets, and RNA-binding domains confer subcompartment specificity. Phase separation contributes to the formation of a multilayer structure of the nucleolus, which determines its functions in ribosome biogenesis [37].

Ribosome biogenesis is far from the only function of the nucleolus. In 1999, the nucleolar retention of proteins was first hypothesized, a phenomenon originally called “nucleolar sequestration,” which describes the ability of the nucleolus to sequester regulatory proteins in response to cellular signals [38]. It was found that in the budding yeast *Saccharomyces cerevisiae*, the cell cycle regulator phosphatase CDC14, which binds to the CFI1/NET1 protein, is sequestered in the nucleolus during the G1 and S phases. It is released during the anaphase, thereby facilitating the cell’s exit from mitosis [36,39,40]. In mammalian cells, it was shown that E3 ubiquitin-protein ligase MDM2 (double minute 2 protein), a p53 inhibitor, can be temporarily localized in the nucleolus, thereby affecting the course of the cell cycle [41]. Notably, many proteins sequestered in the nucleolus are immobile. This has been demonstrated for CDC14 [42], MDM2, DNA (cytosine-5)-methyltransferase 1 (DNMT1) [43], von Hippel-Lindau disease tumor suppressor (VHL), Piwi [40], and E3 ubiquitin-protein ligase RNF8 (RING finger protein 8) [44].

### 3.1. Mechanism of Nucleolar Sequestration, Formation of A-Bodies

Environmental stress factors such as heat shock, extracellular acidosis, and hypoxia trigger the expression of lncRNAs from the intergene spacer region of the rDNA cassette [43]. There are approximately 200 copies of rDNA per haploid genome in mammals. As a rule, genes are distributed over several chromosomes and arranged in a tandem head-to-tail array, each of which contains a transcribed region that produces full-length pre-rRNA and an intergenic spacer (IGS) region [45], which is a bundle of long non-coding transcripts [46,47].

Surprisingly, differences in the structure and length of IGS were found not only between species but also between individuals of the same species [47,48,49]. IGS size ranges from ~2 kbp in yeast up to ~40 kbp in humans. This change in sequence, clearly observed in the increase in size, suggests that regulatory elements or even functional genes may have arisen in these spacer regions during evolution [47]. The human IGS contains many different types of repeats, including simple repeats, microsatellites, repeats derived from retrotransposons, etc. [47,50]. In particular, numerous simple repeats [TC]n or [GA]n are found in human IGS. Three regions within IGS, enriched in simple repeats and predicted to form no secondary structure, are transcribed by Pol I RNA in response to certain cellular stresses and further processed into lncRNA ~300 bp in length [43]. These regions are located at ~16 kb, ~22 kb, and ~28 kb downstream of the rRNA transcription start site. These lncRNAs have been named rIGS16, rIGS22, and rIGS28, respectively [43,51] and are induced by specific cellular stresses; rIGS16 and rIGS22 are induced by heat shock, while rIGS 28 is induced under conditions of extracellular acidosis. 

These lncRNAs play a vital role in the reorganization of the nucleolus, in particular by redirecting certain proteins to this organelle. Such proteins have similar motifs called the amyloid-converting motif (ACM). The ACMs of the studied proteins [43,51,52] contain two distinct domains: the R/H-rich sequence flanking the highly amyloidogenic region. The endogenous rIGSRNA interacts with the R/H residues, allowing the amyloidogenic domains to reach a sufficient concentration, thus triggering the initial fibrillation event followed by protein polymerization. Probably due to the lack of a secondary structure, rIGSRNA has a larger surface area, which provides more binding sites for amyloidogenic proteins. As a result, the formation of an amyloidogenic riboprotein structure, the A-body, begins [43]. As the A-body matures, more proteins aggregate in the nucleolus, which are immobilized through amyloidogenesis. Eventually, these ACM-containing proteins form a reversible mature A-body in the nucleolus (Figure 3).

During the maturation of these organelles, there appears to be an intermediate liquid-like phase that gradually solidifies [50,53,54]. The dismantling of A-bodies occurs within 1–2 h after the end of the stimulus; this process requires the presence of heat shock proteins HSP70 and HSP90 [51,52]. Therefore, A-bodies are amyloid-like entities formed as a result of a phase transition [55]. Other amyloid-like structures are exemplified by the Balbiani bodies observed in *Xenopus* [56] and the pH-controlled transition of the cytoplasm from a liquid to a solid form in yeast [57]. The fibrous, amyloid-like structure of A-bodies distinguishes them from other biomolecular condensates that have liquid-like properties, such as stress granules, nucleoli, and paraspeckles, because liquid-like condensates are dynamic, their constituents are mobile, they do not form fibers detectable by EM, and they usually do not stain with amyloidophilic dyes [31,41,58]. Thus, A-bodies represent a unique nucleolar structure based on their fibrous properties and rIGSRNA dependence.

In the article [59], the protein composition of A-bodies was studied. In the course of the study, data were obtained that showed a high variability in the proteomic composition of A-bodies depending on stress exposure (heat shock, acidosis, or transcriptional/proteotoxic stress). Structures formed as a result of exposure to various stress factors can be different subtypes of A-bodies. Considering that transcription, translation, and metabolic factors are recruited into A-bodies induced by acidosis and heat shock, it can be assumed that gene expression is thus fine-tuned to help cope with specific damage.

At present, there is no unequivocal answer to the question of how proteins are specifically recruited under stress and converted into an amyloid form, although it is obvious that all proteins must have amyloidogenic properties. The model of A-body formation, in which organelle maturation is based on charge-based interactions between RNA and target proteins [43,52], does not take into account the stress-specificity of the proteomic composition. Probably, a second level of regulation of the formation of A-bodies is required. Interestingly, acidosis prior to or co-existing with heat shock prevented the sequestration of Flap Endonuclease 1 (FEN1, specific for heat shock) under conditions of heat shock [59]. If heat alone was sufficient to refold FEN1, the protein would be recruited into A-bodies. Probably, acidotic conditions block the elements of the heat shock signaling pathway, i.e., the expression of heat shock-induced transcripts responsible for the sequestration of proteins in A-bodies is suppressed. It is possible that stress-specific targets can be modified in a certain way (for example, post-translationally), which leads to a phase transition and the formation of physiological amyloid aggregates. The resolution of this puzzle may, to some extent, provide insight into the cellular response to stress and potential sites of dysregulation for pathological aggregation.

It has been shown that only ~20% of heat shock A-body components can be found in acidosis-induced A-bodies [59]. Proteins such as VHL and CDC73 are characteristic of all types of A-bodies and can be used as A-body markers in research.

#### 3.1.1. VHL

The von Hippel–Lindau tumor suppressor protein (VHL) is a vital component of the VBC-Cul2 E3 ubiquitin ligase complex as it acts as a substrate recognition protein and mediates the specificity of the degradation process [60,61,62,63]. VHL promotes recruitment, ubiquitination, and subsequent proteasomal degradation of the alpha subunit of hypoxia-inducible factor (HIF) [64,65]. A decrease in extracellular pH (acidosis) triggers the translocation of VHL into the nucleolus, neutralizing its ability to degrade nuclear HIF even in the presence of oxygen. It has been shown that VHL passes from a highly dynamic nuclear-cytoplasmic distribution under standard growth conditions (21% O_2_, pH 7.4) to an immobilized state in the nucleolus when exposed to extracellular acidosis (1% O_2_, pH 6.0) [66]. Only upon normalization of extracellular pH, VHL is released from the nucleolus and restores its mobility [66]. VHL mapping revealed an amino acid fragment, originally called the nucleolar retention signal (NoDS), which is thought to be necessary and sufficient for protein immobilization in the nucleolus [44]. 

A study [51] analyzed the VHL fragment (residues 104–140; VHL104–140) and found that it interacts more effectively with endogenous rIGS 28 RNA during acidosis compared to other amyloidogenic or non-amyloidogenic regions. A more thorough study of VHL104–140 revealed the presence of an arginine/histidine cluster (residues 104–121), which is activated after a decrease in extracellular pH, mediating the retention of VHL in the nucleolus. This arginine/histidine-rich sequence flanks a highly amyloidogenic motif (residues 122–140). It was found that proteins containing these motifs have all the properties of amyloids [30]. Subsequently, the term “NoDS” was replaced by “amyloid-converting motif” (ACM) to emphasize its role in the transformation of the nucleolus into A-bodies. Notably, ACM is inactivated upon return to neutral pH conditions. Further research is needed to elucidate the mechanisms by which extracellular hydrogen ions activate ACM VHL, as well as what drives VHL sequestration during heat shock.

#### 3.1.2. CDC73

Cell division control protein 73 (CDC73) is one of the key cellular regulators and is involved in transcriptional and post-transcriptional control pathways. It may influence cell cycle progression through the regulation of cyclin D1/PRAD1 expression. The protein is a component of the RNA polymerase II-associated protein 1 (PAF1) complex (PAF1C), which performs many functions during transcription by RNA polymerase II and is involved in the regulation of the development and maintenance of pluripotency of embryonic stem cells. Since CDC73 is an important marker protein present in all types of A-bodies, from flies to humans [67], its use as a marker in research provides a further opportunity to study the formation of these structures in various cell types.

A study [67] showed that the formation of A-bodies can occur in cells derived from various species, such as monkeys, mice, rats, chickens, fish, and flies. A-bodies were also detected in *D. melanogaster* (flies) and *S. cerevisiae* (yeast). It is important that the transcriptome sequences of scaffold lncRNAs are not conserved in humans, mice, or chickens [67]. There is a general pattern: stress leads to the expression of lncRNA of low complexity from the region immediately below the rRNA coding sequence in all three organisms [67,68]. These data suggest that the genomic arrangement and function of these scaffold transcripts are conserved across species, despite the lack of conservation in their primary sequence.

### 3.2. Functions of A-Bodies

Although A-bodies have structural similarities to pathological amyloids, they are physiological and serve biological functions. Many proteins sequestered in A-bodies, for example, CDK1, POLD1, and DNMT1, are involved in the regulation of the cell cycle [51]. The capture and temporary retention of such proteins in A-bodies mediate the suppression of metabolic activity [69] and the arrest of DNA synthesis [51]. This state is different from the resting state (G0) in the cell cycle when cells stop proliferating but remain metabolically active. Therefore, the biological role of A-body formation might be to provide cellular dormancy as an adaptive response to environmental stressors. That is, it is the “solid-like” properties of A-bodies that lead to a loss of metabolic activity and contribute to the transition of cells to a sleep mode.

Another function of A-bodies is the local synthesis of nuclear proteins [70]. A study [49] identified translating ribosomes organized along filamentous assemblies that characterize amyloid bodies. rIGS RNA silencing disrupted nuclear translation in acidosis and heat shock. Nuclear translation cessation correlated with A-body disassembly during stress recovery. Thus, it is likely that stressful conditions such as heat shock and acidosis stimulate nuclear translation in A-bodies. Interestingly, lower eukaryotes (flies and yeasts) lacked local nuclear protein synthesis [67]. This is most likely due to the fact that A-body-mediated translation is a later evolutionary feature of this structure.

Therefore, A-bodies function not only as a preserving structure but also play an adaptive role in the fight against stress.

## 4. Conclusions

In this review, we attempted to summarize and analyze the literature data on two non-coding RNA-containing stress-induced nuclear MLOs that are formed by a similar principle and have an overlapping protein composition but significantly different physical properties, morphology, and functional activity. It could be concluded that the amyloid-like structure of A-bodies correlates with the more “passive” role of A-bodies in cellular stress response, including the storage of cellular protein material, while dynamic properties of nSBs correspond to the active stress-induced change of expression programs by these MLOs. In any case, a further coordinated study of these two LLPS-mediated nuclear compartments will expand the scientific knowledge about the processes of formation of stress-inducible membrane-less organelles before their participation in the regulation of cell signaling pathways, and bring us closer to understanding the basic role of non-coding RNA in the regulation of the stress response.

## Figures and Tables

**Figure 1 ijms-24-08108-f001:**
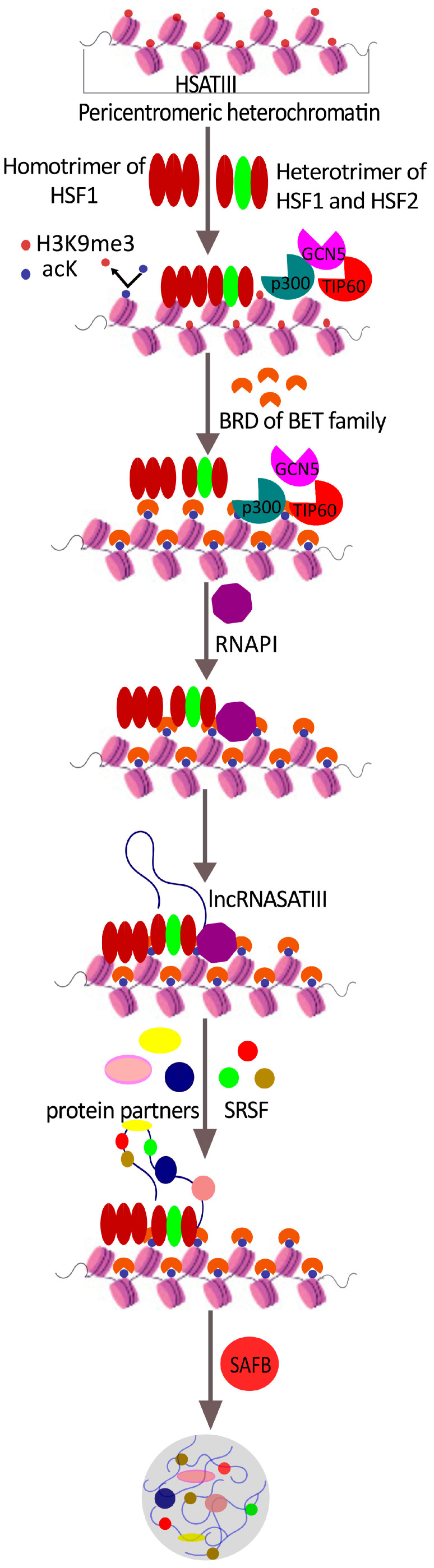
The proposed mechanism of the nSB biogenesis. As a result of heat shock, transcription of highly repetitive satellite III architectural noncoding RNAs is initiated. Subsequently, HSF1 is homotrimerized, as well as heterotrimerized with HSF2, and recruited to the pericentromeric heterochromatin region. Subsequent recruitment of histone acetyltransferases HATs (GCN5, TIP60, and p300) results in redistribution of histone acetylation, recruitment of the members of the BET family of bromodomain-containing (BRD) proteins, and chromatin remodeling. DNA transcription is activated by RNAP II. The formation and maturation of nSB proceed with the involvement of signal recognition particle (SRP) and other protein partners. Scaffold attachment factor B1 (SAFB, which is one of the hnRNPs) binds to already formed nSBs.

**Figure 2 ijms-24-08108-f002:**
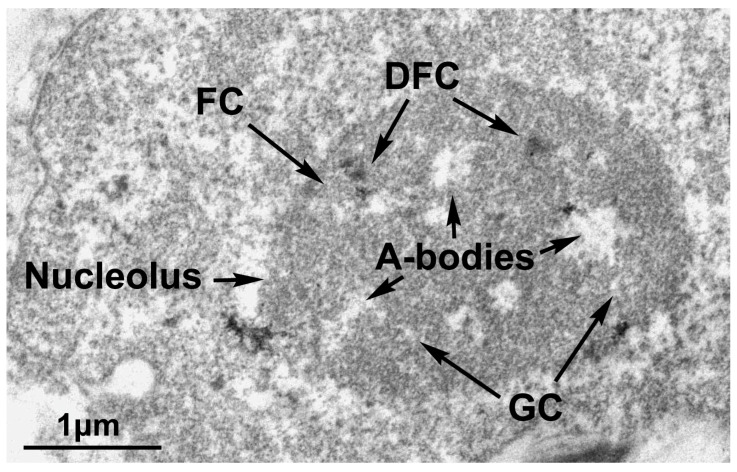
The structure of the nucleolus with A-bodies. The transmission electron micrograph shows the nucleus of a MCF7 cell, where the arrows indicate the nucleolus with the subnucleolar compartments defined by their differential stainings: FC, fibrillar center; DFC, dense fibrillar components; and GC, granular components. A-bodies are stress granules that appear after 1 h of treatment at 43 °C; they have fibrillar structures inside them. The scale bar corresponds to 1 μm.

**Figure 3 ijms-24-08108-f003:**
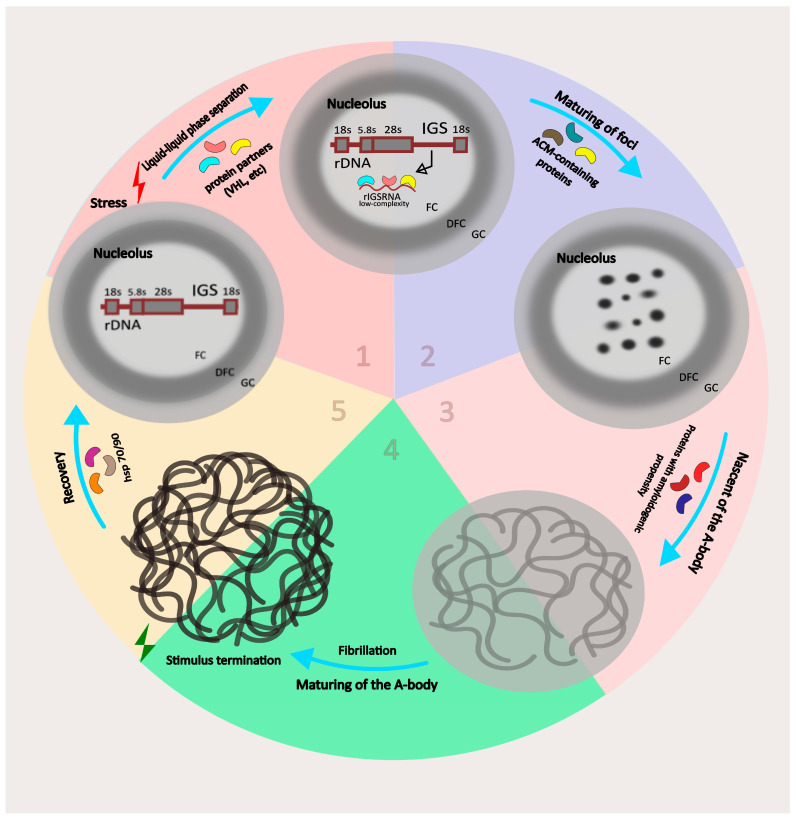
Stress induces transcription of low-complexity ribosomal intergenic spacer RNA (rIGSRNA) in the nucleolus. At stage 1, rIGSRNA interacts with amyloidogenic proteins and the LLPS process starts. At stage 2, liquid-like amyloidogenic “foci” are formed, consisting of rIGSRNA and amyloidogenic proteins. At stage 3, the fibrillar solid A-body matures. At stage 4, a mature A-body is formed. Mature A-bodies reversibly accumulate a large amount of proteins and promote cellular rest in response to stress. After stress relief, HSP-mediated disassembly of the A-body occurs (step 5).

## Data Availability

No new data were created or analyzed in this study. Data sharing is not applicable to this article.

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
