# Peer review of "On the Roles of the Nuclear Non-Coding RNA-Dependent Membrane-Less Organelles in the Cellular Stress Response"

_ijms, 2023, doi:10.3390/ijms24098108_

Round 1

Reviewer 1 Report

This is a review paper on the current knowledge of formation of two
main kinds of nuclear RNA-dependent membraneless organelles (MLOs) as
a response to cellular stress, nucleolar stress bodies (nSBs) and
amyloid bodies (ABs).  This review is mostly descriptive and concludes
with a short comparison between the two types of MLOs.

Here are my recommendations for improving this review:

(1) There are too many abbreviations that are not used in a consistent
way.  For example, LLPS and MLOs appears in the abstract without being
defined.  I recommend not to introduce an abbreviation if it is not
used at least three times in the text.

(2) The first sentence of Introduction has typos.

(3) It is unclear what the first paragraph on page 3, starting with
"Activation of pericentric ..", is all about.  Are the authors
describing a hypothesis or experimental observations?  If the latter,
which experiments provide support to the described process?

(4) The quality of Figure 1 is below the publication quality bar.
Details cannot be seen. The font of the text is too small. Redo this
figure and scale it appropriately.

(5) Provide additional figure referring to the layered structure of
nucleolus and stress granules observed under the electron microscope
(section A-bodies on page 6, the first paragraph).

Author Response

This is a review paper on the current knowledge of formation of two main kinds of nuclear RNA-dependent membraneless organelles (MLOs) as a response to cellular stress, nucleolar stress bodies (nSBs) and amyloid bodies (ABs). This review is mostly descriptive and concludes with a short comparison between the two types of MLOs.

Response: We are grateful to this Reviewer for careful reading of our work and providing valuable comments.

Here are my recommendations for improving this review:

(1) There are too many abbreviations that are not used in a consistent
way. For example, LLPS and MLOs appears in the abstract without being
defined. I recommend not to introduce an abbreviation if it is not
used at least three times in the text.

Response: We removed some abbreviation

(2) The first sentence of Introduction has typos.

Response: The typos were corrected

(3) It is unclear what the first paragraph on page 3, starting with "Activation of pericentric ..", is all about. Are the authors describing a hypothesis or experimental observations? If the latter, which experiments provide support to the described process?

Response: Activation of pericentric heterochromatin at the 9q12 locus accompanied with HSAT III accumulation was shown in work [15] by FISH and immunofluorescence analysis.

(4) The quality of Figure 1 is below the publication quality bar. Details cannot be seen. The font of the text is too small. Redo this figure and scale it appropriately.

Response: We upgraded this figure

(5) Provide additional figure referring to the layered structure of nucleolus and stress granules observed under the electron microscope (section A-bodies on page 6, the first paragraph).

Response: We added figure with TEM picture of stressed nucleolus

Reviewer 2 Report

s

Author Response

This paper is a review on stress-induced bodies in cells, focussing on stress bodies and A-bodies, both of which are formed from non-coding RNA and (mainly intrinsically disordered) proteins in a process of liquid-liquid phase separation. This is a detailed and well-researched review, that makes a useful contribution to the literature. It is well written and presents a coherent account. It could be improved by a number of corrections, as set out below.

Response: We are grateful to this Reviewer for careful reading of our work and providing valuable comments.

  1. Lines 18, 66, 254. Through most of the review, the two aggregates discussed in detail are
    usually described as nuclear stress bodies and (nucleolar?) A-bodies. However on line 18 they are described differently, and in a different order. Please use consistent nomenclature. Line 66 (heading) should presumably be nuclear.

Response: Corrected

  1. Line 27 - delete according

Response: Corrected

  1. Line 41 – Archaea

Response: Corrected

  1. Line 57 - it would be helpful to give some detail on how fast these structures form and disappear

Response: We improved this sentence

  1. Line 62 - mechanism

Response: Corrected

  1. Figure 1 has very small text which needs making bigger. The word heterochromatin has become messed up. Please check that all text is described in the text or the figure legend -H3K9me3 is not mentioned, and I suspect SRF should be SRP.

Response: Corrected

  1. Lines 117-118 change to while lower (39-40 C) or higher (45C) temperatures

Response: Corrected

  1. Line 119 remove the (in the hyperosmotic stress)

Response: Corrected

  1. Lines 132-134. This sentence does not make sense and needs rewording. Maybe the word with needs replacing by have?

Response: We rewrote this sentence

  1. Line 153 factor

Response: Corrected

  1. Line 210 has

Response: Corrected

  1. Line 214 remove the comma after above

Response: Corrected

  1. Line 273 should internally be intrinsically?

Response: Corrected

  1. Lines 280-281 Saccharomyces cerevisiae should be in italics

Response: Corrected

  1. Line 302 in humans

Response: Corrected

  1. Line 217 should be something like: a R/H-rich sequence, and an adjacent highly
    amyloidogenic region.

Response: Corrected

  1. Figure 2 - some of the text here is also very small

Response: Corrected

  1. Line 337. Please explain in more detail what you mean by gel-like solid-like

Response: We rewrote this sentence

  1. Line 370 A-body

Response: Corrected

  1. Line 371 remove the comma after Proteins

Response: Corrected

  1. Lines 382 and 383 - the 2 should be subscript

Response: Corrected

  1. Line 412. Please explain what you mean by body-specific conditions

Response: It means heat shock and hypoxia conditions. We rewrote this sentence

  1. Line 416 Please use a different word to replace below - this does not make sense.

Response: We rewrote this sentence

  1. Lines 429-432: this is an important sentence, so please re-write (particularly the second half)
    to explain more clearly what you mean.

Response: We rewrote this sentence

  1. Line 436 - A-body

Response: Corrected

  1. Line 440: A-body-mediated

Response: Corrected

  1. Line 447 delete carefully

Response: Corrected

  1. Lines 447-450. It would be useful to reword this sentence and make it longer, to explain the
    meaning more clearly.

Response: We rewrote this sentence

  1. Line 451. Explain more clearly what you mean by similar and simultaneously different

Response: We rewrote this sentence

  1. Line 453. Insert and before their participation

Response: Corrected

  1. Ref 1: The origin of life

Response: Corrected

  1. Ref 8 C elegans should be in italic

Response: Corrected

  1. Refs 58 and 59 are identical

Response: Corrected

  1. Ref 71 needs page numbers

Response: Corrected

Reviewer 3 Report

The authors have provided a nice review of two stress-induced nuclear membraneless organelles.  The manuscript is well written and could be of broad interest to researchers focusing on membraneless organelles, liquid-liquid phase separation, functional amyloids and intrinsically disordered proteins in biophysical and biomedical science.  I include my minor comments below.

The authors mentioned LLPS of IDPs in the abstract as the mechanism of forming MLOs.  However, LLPS was not the focus when discussing each protein separately and was included more as summary paragraphs when discuss nSBs.  It might help by providing a little bit the relevance of LLPS and/or IDPs for each system the authors introduced if there exists some literatures.  For instance hnRNP, RS domain, RNAPII and many other proteins the authors mentioned have been studied for their LLPS and/or co-phase separation with RNA.  

The terminology IDP has only been mentioned in the abstract.  It is difficult for the reader to understand the role of intrinsically disorder in LLPS and formation of nSBs.  

When discussing nSBs and A-bodies, a few terms about the states are used such as fluid-, gel-, solid-, and amyloid-like.  Sometimes two of them are used together (P8 L337, gel-like solid-like).  It could be confusing to the readers considering the definition of these states.  A few sentences to explain the way (theoretical or experimental) of interpreting these states might be helpful.

The same comment about role of IDPs in nSBs applies to A-bodies.  Studies about the role of disordered region in these proteins if existed can be introduced.  

The reader might also be curious about the mechanisms of LLPS and aggregation underlying these functional amyloids in contrast to the pathological amyloids which are usually irreversible.

P1: LLPS and MLOs are used the first time they appear in the abstract.

Should "A-bodies" be section 3 and conclusion be section 4?

Author Response

The authors have provided a nice review of two stress-induced nuclear membraneless organelles. The manuscript is well written and could be of broad interest to researchers focusing on membraneless organelles, liquid-liquid phase separation, functional amyloids and intrinsically disordered proteins in biophysical and biomedical science. I include my minor comments below.

Response: We are grateful to this Reviewer for careful reading of our work and providing valuable comments.

The authors mentioned LLPS of IDPs in the abstract as the mechanism of forming MLOs. However, LLPS was not the focus when discussing each protein separately and was included more as summary paragraphs when discuss nSBs. It might help by providing a little bit the relevance of LLPS and/or IDPs for each system the authors introduced if there exists some literatures. For instance hnRNP, RS domain, RNAPII and many other proteins the authors mentioned have been studied for their LLPS and/or co-phase separation with RNA.

Response: We slightly rewrote our work in this context. Unfortunately, available literature on this subject is very scarce.

The terminology IDP has only been mentioned in the abstract. It is difficult for the reader to understand the role of intrinsically disorder in LLPS and formation of nSBs.

Response: Corrected

When discussing nSBs and A-bodies, a few terms about the states are used such as fluid-, gel-, solid-, and amyloid-like. Sometimes two of them are used together (P8 L337, gel-like solid-like). It could be confusing to the readers considering the definition of these states. A few sentences to explain the way (theoretical or experimental) of interpreting these states might be helpful.

Response: Corrected

The same comment about role of IDPs in nSBs applies to A-bodies. Studies about the role of disordered region in these proteins if existed can be introduced.

Response: Unfortunately, there are no studies dedicated to the analysis of the roles of IDPs in A-bodies.

The reader might also be curious about the mechanisms of LLPS and aggregation underlying these functional amyloids in contrast to the pathological amyloids which are usually irreversible.

Response: It is very debatable topic and we would not like to delve into this without reliable experimental data

P1: LLPS and MLOs are used the first time they appear in the abstract.

Response: Corrected

Should "A-bodies" be section 3 and conclusion be section 4?

Response: Corrected